# Facile Production of a Fenton-Like Photocatalyst by Two-Step Calcination with a Broad pH Adaptability

**DOI:** 10.3390/nano10040676

**Published:** 2020-04-03

**Authors:** Siyang Ji, Yanling Yang, Xing Li, Hang Liu, Zhiwei Zhou

**Affiliations:** College of Architecture and Civil engineering, Beijing University of Technology, No.100 Xi Da Wang Road, Chao Yang District, Beijing 100124, China; jsy_3021826@163.com (S.J.); yangyanling@bjut.edu.cn (Y.Y.); lixing@bjut.edu.cn (X.L.); liuhang98596@163.com (H.L.)

**Keywords:** Fenton-like photocatalysis, graphitic carbon nitride, two-step calcination, composite materials

## Abstract

A novel heterogeneous Fenton-like photocatalyst, Fe-doped graphitic carbon nitride (Fe-g-C_3_N_4_), was produced by facile two-step calcination method. This Fe–g–C_3_N_4_ catalyzed rhodamine B degradation in the presence of H_2_O_2_ accompanied with visible light irradiation. transmission electron microscopy(TEM), x-ray diffraction (XRD), FT-IR, x-ray photoelectron spectroscopy (XPS), and photoluminescence fluorescent spectrometer (PL) characterization analysis methods were adopted to evaluate the physicochemical property of samples. It can be observed that the Fe-g-C_3_N_4_ exhibited excellent photocatalytic Fenton-like activity at a wide pH range of 3–9, with rhodamine B(RhB) degradation efficiency up to 95.5% after irradiation for 45 min in the presence of 1.0 mM H_2_O_2_. Its high activity was ascribed to the formation of Fe–N ligands in the triazine rings that accelerated electron movement driving the Fe(III)/Fe(II) redox cycle, and inhibited photo-generated electron hole re-combinations for continuous generation of reactive oxygen species by reactions between Fe(II) and H_2_O_2_. The main active oxygen species were hydroxyl radicals, followed by superoxide radicals and hole electrons. This produced catalyst of Fe–g–C_3_N_4_ shows excellent reusability and stability, and can be a promising candidate for decontamination of wastewater.

## 1. Introduction

Advanced oxidation process such as the Fenton reaction and photocatalysis has been extensively studied due to their high degradation activity toward the refractory organic pollutants in recent years [1]. The Fenton reaction has been proven to mineralize most of the organic pollutants [2]. However, the conventional homogeneous Fenton process suffers from the narrow pH range and large amounts of iron precipitation sludge [3]. To overcome these shortcomings, researchers worldwide have focused on the heterogeneous Fenton reaction that immobilizes iron species on the carriers such as clays [4], zeolites [5], and photocatalytic compounds. Furthermore, introducing the Fe–N ligands can effectively increase the pH range of the conventional Fenton reaction [6]. Ligands such as phthalocyanines and tetra-amido macrocyclic can form Fe–N ligands. For example, Gupta et al. used the Fe–TAML catalysts to degrade pentachlorophenol at a high pH of 10 [7]. Zhu et al. developed a Fe(II)-phthalocyanine compound for the degradation of antibiotic carbamazepine [8]. The formation of Fe–N ligands not only stabilizes the iron species, but also accelerates the Fe(III)/Fe(II) redox cycle, thus resulting in the ability to work at higher pH values [9].

Graphitic carbon nitride (g–C_3_N_4_) is a polymer that combines superior photocatalytic properties with chemical stability [10]. It is easy to prepare through the one-step polymerization of cheap precursors of melamine, cyanoguanidine, and urea [11], and it contains structures with six nitrogen lone-pair electrons, which are ideal sites for chemical modification. However, the fast electron-hole recombination limited the application of one-step g–C_3_N_4_. Common modification methods are thermal oxidation exfoliation and surface modification with other ions [12]. On one hand, the two-step calcination (thermal oxidation exfoliation) could potentially improve the photocatalytic activity of one-step g–C_3_N_4_ by increasing specific surface area and electron transport ability [13]. On the other hand, g–C_3_N_4_ doping with metal atoms such as Co, Fe, Cu, and Pb can enhance photocatalytic and oxygen-reduction activity. Wang et al. [14] used a one-step calcination method to prepare CoS_2_/g–C_3_N_4_–rGO for reduction of Cr (VI). Guo et al. [15] used one-step calcination method to synthesize a novel Fe_2_O_3_@g–C_3_N_4_ for the degradation of tetracycline. Dong et al. [16] developed Cu–g–C_3_N_4_ by a modified impregnation method to degrade rhodamine B (RhB). Under visible light irradiation in the presence of H_2_O_2_, the Fenton-like photocatalysis of the above multifunctional compounds will take place, accompanying photoelectron-hole pairs that are generated and active radicals are produced that can degrade organic pollutants [17], meanwhile synergistic effects result from a combination with the Fenton reaction [18]. Specifically, photocatalysis can be combined with Fenton-like oxidation technology, on one hand, photo-generated electrons can promote the reduction of Fe^3+^; on the other hand, H_2_O_2_ in the Fenton reaction reacting with photo-generated electrons can reduce the electrons-hole recombination probability, thereby improving photocatalytic degradation efficiency [19,20].

Herein, we synthesized a Fenton-like photocatalyst by two-step calcination that doped Fe upon g–C_3_N_4._ The Fe–g–C_3_N_4_ catalysts were systematically characterized. Synergistic degradation of RhB by Fenton-like photocatalysis was investigated. The high pH adaptation from 3 to 9 was assessed, and the main active species and mechanism of reaction were clarified.

## 2. Materials and Methods

### 2.1. Materials

All chemical reagents used in these experiments were at least of analytical grade. Urea (CH_4_N_2_O, ≥98.5%), iron chloride hexahydrate (FeCl_3_·6H_2_O, 99.0%), isopropyl alcohol (IPA, ≥99.5%), ethylenediaminetetraacetic acid disodium salt (EDTA-2Na, 99.0%), and RhB were purchased from Aladdin Industrial Corporation (Shanghai, China). All chemicals were used as received without further purification. Stock solutions were prepared with Milli-Q ultrapure water.

### 2.2. Preparation of the Catalysts

Urea and FeCl_3_•7H_2_O were used for the preparation of the calcination of g–C_3_N_4_ and Fe–g–C_3_N_4_.The two-step calcination of g–C_3_N_4_ was optimized according to the method described in a previous study with modifications [21]. One-step calcination (product with prefix ‘1^st^’) was performed by heating 5 g urea with or without 0.105 g FeCl_3_•7H_2_O to 550 °C at 5 °C/min for 4 h in a muffle furnace. For two-step calcination (‘2^nd^’), the calcinated products were allowed to spontaneously cool to room temperature, followed by a 2 h-calcination step in a tube furnace at 520 °C (5 °C/min) in a nitrogen atmosphere (Figure 1a).

### 2.3. Characterization

The morphology of the products was characterized by field-emission scanning electron microscopy (FE-SEM, SU-8020, Hitachi, Tokyo, Japan) and high-resolution transmission electron microscopy (FEI, Tecnai G2 F20, USA). Their crystal structure was determined by x-ray diffraction (XRD-7000, Shimadzu, Kyoto, Japan) using CuKα radiation with 2θ set from 10 to 80°. Fourier transform infrared (FTIR) was recorded with a NICOLET iS10 FTIR spectrometer. Surface chemical composition was analyzed by x-ray photoelectron spectroscopy (XPS, Thermo Fisher Scientific, Waltham, USA) with a monochromatic Al Kα source (1486.6 eV). All binding energies were calibrated based on the C1s peak at 284.8 eV. The specific surface area was measured by nitrogen adsorption–desorption isotherms at 77 K using the Barret–Joyner–Hallender method (NOVA 3200e Sorptometer, Quantachrome, Florida, USA). The optical properties of the catalysts were analyzed by a photoluminescence fluorescent spectrometer (PL, FLS980, Livingston, Edinburgh) and an ultraviolet-visible diffuse reflectance spectrometer (UV–Vis DRS; 3600 plus, Shimadzu, Kyoto, Japan), respectively. Presence of free radicals was confirmed by electron paramagnetic resonance (EPR) (JES-FA200 JEOL, Tokyo, Japan) under visible light irradiation using 5,5-Dimethyl-1-pyrroline N-oxide (DMPO) as a trapping compound.

### 2.4. Degradation Performance of the Catalysts

The catalytic activity of the obtained catalysts were evaluated by RhB degradation efficiency at room temperature. For photocatalysis or Fenton-like photocatalysis, a 300W Xe lamp (CEAULIGHT, CEL-HXF300E7,Beijing, China) equipped with a 400 nm UV filter was used that emitted visible light (400–780 nm) at 280 mW/cm^2^. RhB was used as the target pollutant at an initial concentration of 10 mg/L (filtered through 0.45-μm acetate membranes prior to use).

For the photocatalysis reaction, as a typical process, 0.02 g of the catalyst (g-C_3_H_4_ and Fe–g–C_3_H_4_) was added into 100 mL RhB solution (10 mg/L). Before irradiation, the suspension was stirred for 40 min in the dark to ensure light-independent adsorption–desorption equilibriums had been reached (presented in Appendix A in Appendix A). Then, a 300 W Xe lamp equipped with a cut off filter (>420 nm) as a light source and placed 12 cm above the surface of the reaction solution was turned on.

For the Fenton-like photocatalysis reaction, under the same condition as the photocatalysis reaction, except that 1.0 mmol/L H_2_O_2_ was added into the suspension before the irradiation, the doses of which were determined by our pre-experiments (Appendix A). A sample without any catalyst control was included as the control. At a specific time interval, 3 mL of the suspension solution was collected and filtered through a 0.22 μm filter for measure by a UV–Vis Spectrophotometer (UV-2600, SOPTOP, China) at 554 nm. When studying the influence of pH, 0.1 mol/L HCl and NaOH solutions were employed to adjust the initial pH value. The influence of pH on the treated solution during the reaction was not addressed, since the fluctuations of pH value were small (±0.2) [22].

The degradation kinetics of RhB in photocatalysis or the Fenton-like photocatalysis process were fitted based on a pseudo-first-order kinetic model (Equation (1)), where C_0_ (mg/L) is the concentration of RhB after adsorption in dark (mg/L), Ct is the concentration at time t (min), and k (min^−1^) is the pseudo first-order rate constant.
lnC_t_/C_0_ = −kt(1)


The stability and reusability of the synthesized catalysts were tested by repeating the Fenton-like photocatalysis experiment. For this, at the end of a photocatalytic experiment, the used Fe–g–C_3_N_4_ was filtrated, washed, and dried before being used for the following test.

## 3. Results

### 3.1. Characterization

TEM images of the two calcination products were similar, but the material turned from flaxen-collared to brownish upon Fe doping (Figure 1b,c). Element mapping of Fe–g–C_3_N_4_ (Figure 1d–g) confirmed the presence of Fe responsible for the color change. The Fe-signal overlapped with that of N, indicating interactions between Fe and N. One-step or two-step calcination of Fe–g–C_3_N_4_ resulted in a nanosheet structure similar to that of g–C_3_N_4_ visible by SEM (Appendix A). The S_BET_ of 1^st^ g–C_3_N_4_, 2^nd^ g–C_3_N_4_, 1^st^ Fe–g–C_3_N_4_ and 2^nd^ Fe–g–C_3_N_4_ was 56.78, 78.54, 45.61, and 63.52 m^2^/g, respectively, which implies that the two-step calcination can increase pore diameter and the BET specific surface area of both materials (Appendix A). The higher surface area and larger pore diameter offer more active sites at the catalyst surface, which potentially enhance catalytic activity.

XRD patterns of materials by one-step and two-step calcination (Figure 2a) displayed a peak of the (002) plane around 27.5°, characteristic of a graphite-like structure (JCPDS No. 87-1526). The peak at 13.0° was attributed to the crystal plane of (100) in-planar ordering of tri-s-triazine units [23], and this peak disappeared following Fe-doping, while the peak intensity for the (002) plane was significantly reduced, suggesting the absence of a rigid layer. There were no extra diffraction peaks observed for other phases such as Fe-related secondary phase or impurity, indicating Fe successfully doped into the g–C_3_N_4_ lattice. The FTIR spectra (Figure 2b) displayed characteristic bands in all samples for stretching vibration of –NH and –NH_2_ groups (3000–3400 cm^−1^), C=N, and C–N bonds (1240–1640 cm^−1^) [24], and triazine (805 cm^−1^). Iron peaks were not detected in Fe–g–C_3_N_4_, suggesting that all Fe was doped into the g–C_3_N_4_.

As shown in Figure 2c, the PL emission of all the samples exhibited a primary emission peak around 450 nm, which is consistent with the earlier reports [25,26,27]. We found that two-step calcination could reduce peak intensity. As Fe doping strongly reduced peak intensity, probably due to the transfer of photoelectrons to Fe through chemical bridging, resulting in inhibited recombination between photoelectrons and holes [28], this was of benefit to the photocatalytic activity of the catalyst activity of the catalyst. PL emission spectra are attributed to the band-gap emission in light of its absorption band edge, estimated by the UV–Vis DRS (Appendix A).

The surface chemical state was analyzed by XPS (Figure 2d), and the atomic content was calculated (Appendix A). The proportion of Fe element was increased by two-step calcination compared to the one-step products. Peaks in the high-resolution of the N1s and Fe2p spectra were attributed to molecular components. The N1s spectrum of g–C_3_N_4_ in Figure 2e contains four peaks that were attributed to pyridinic N (395.3 and 397.6 eV) [29], C–N (396.6 eV), and C=N–C (401.1 eV), respectively [30]. Fe-doping added peaks at 397.35 and 398.0 eV corresponding to iron nitride (N–Fe) and (Fe–(CN)), respectively [31], accounting for 4.8 and 4.5% of the total N. The Fe2p spectrum in Figure 2g illustrates that surface Fe represents Fe bonded with nitrogen (Fe–N) at 706.7 eV, FeO at 709.3 eV, and Fe_2_O_3_ at 724.0 eV, comprising 37.8%, 36.9%, and 6.4%, respectively [32,33]. No metallic Fe was detected on the surface of Fe–g–C_3_N_4_, confirming that all Fe was completely doped into the g–C_3_N_4_ lattice and mainly in the form of Fe–N.

### 3.2. Degradation Performance of the Catalysts

The photocatalysis performance of the catalysts were tested under visible light irradiation. As exhibited in Figure 3a, the degradation efficiency of RhB by photolysis can be ignored in the absence of the catalyst, whereas the photocatalytic degradation efficiency of RhB by 2^nd^ g–C_3_N_4_ was 51.7% within 45 min, which was 13.2% higher than that of 1^st^ g–C_3_N_4_. The photocatalytic activities were poor for both samples doped with Fe (1^st^ Fe–g–C_3_N_4_ and 2^nd^ Fe–g–C_3_N_4_). The sequence of degradation effects from high to low was 2^nd^ g–C_3_N_4_, 1^st^ g–C_3_N_4_, 2^nd^ Fe–g–C_3_N_4_, and 1^st^ Fe–g–C_3_N_4_. From the fitted curve in Figure 3b, we found the degradation process of RhB conformed to the first-order kinetics model, and the calculated rate constant was 0.01446, 0.01077, 0.00436, 0.00776 min^−1^, for 2^nd^ g–C_3_N_4_, 1^st^ g–C_3_N_4_, 2^nd^ Fe–g–C_3_N_4_, and 1^st^ Fe–g–C_3_N_4_, respectively. The photocatalysis performance of 2^nd^ g–C_3_N_4_ was the best, which was ascribed to the highest S_BET_ (78.535 m^2^/g) and pore volume (0.383 cc/g) that had the most active sites. It can be concluded that two-step calcination improved properties in the degradation of dyes.

It can be seen from Figure 3c that the degradation efficiency of RhB with the addition of H_2_O_2_ only reached about 10% at 45 min. The H_2_O_2_ under visible light without a photocatalyst could not improve RhB degradation efficiency, both confirming that H_2_O_2_ only or H_2_O_2_ with visible light irradiation could marginally degrade RhB [34]. The Fenton-like photocatalysis performance under visible light irradiation in the presence of 1.0 mM H_2_O_2_ was investigated. In the presence of H_2_O_2_, the degradation efficiency of RhB greatly increased after doped Fe ions (Figure 3c). The 2^nd^ Fe–g–C_3_N_4_ had the highest degradation efficiency, reaching up to 95.5% within 45 min (89% within 15 min). In comparison, the degradation efficiency of RhB by 2^nd^ Fe–g–C_3_N_4_ under photocatalysis, Fenton-like, and Fenton-like photocatalysis within 45 min was 24.2%, 76.4%, and 95.5%, respectively.

In addition, it can be seen that the 2^nd^ Fe–g–C_3_N_4_ and 1^st^ Fe–g–C_3_N_4_ showed higher catalytic activity than that of the 2^nd^ g–C_3_N_4_ and 1^st^ g–C_3_N_4_. The rate constant of the 2^nd^ Fe–g–C_3_N_4_ (0.5424 min^−1^) was the highest among all the materials, which was about 1.46, 1.86, and 2.32 times higher that of the 1^st^ Fe–g–C_3_N_4_ (0.3722 min^−1^), 2^nd^ g–C_3_N_4_ (0.2921 min^−1^), and 1^st^ g–C_3_N_4_ (0.2335 min^−1^). This phenomenon was likely because iron doping can promote the production of active oxygen species [35].

The effect of pH value was tested with the 2^nd^ Fe–g–C_3_N_4_ only, in the presence of H_2_O_2_ (Figure 4). It shows that at the pH values of 3, 5, 7, and 9, the degradation efficiency of RhB was 96.0%, 95.2%, 92.9%, and 96.6%. Although RhB degradation was more rapid at a pH of 3, degradation was still highly efficient at a pH of 5–9. The reason is that Fe doping into the g–C_3_N_4_ formed Fe–N bonds, which promoted the migration and circulation of Fe ions and avoided the precipitation of iron salts during the Fenton-like reaction. This represents a major improvement compared to the classic Fenton reaction, which requires a pH of 2–4. The Fe–N ligand not only keeps the iron stable under different pH values, but also improves the redox properties [36].

We performed recycling tests by recovering the nanoparticles. As shown in Figure 5a, the remove efficiency of RhB by 2^nd^ Fe–g–C_3_N_4_ had no significant reduction after five cycles, with the degradation efficiency of 95.5%, 94.9%, 93.6%, 92.5%, and 90.3%, respectively. Furthermore, the fresh sample and that used after five times had similar XRD characteristic peaks (Figure 5b). The above results indicate the catalytic had an excellent Fenton-like photocatalytic reusability and stability.

### 3.3. Mechanism of Fenton-Like Photocatalytic Reaction

The reactive species were detected under light and dark conditions by EPR, as shown in Figure 6a,b. Under visible light irradiation, the specific spectrum clearly appeared, which proved the production of •OH and •O_2_^−^ during the catalytic process. In order to further prove the main reactive species, trapping tests were carried out. IPA (1 mmol/L) was used as a scavenger of •OH in the reaction, EDTA-2Na (1 mmol/L) and N_2_ were used as scavengers of holes and•O_2_ [37], respectively. Compared with the residual RhB (~4%) for 45 min without any scavenger, the residual RhB was 36.7, 8.1, and 17.2%with IPA, EDTA-2Na, and N_2_, respectively (Figure 6c). This result indicates that •OH was the primary reactive species, followed by •O_2_^−^ and hole electrons.

Based on the above results and discussion, a tentative mechanism for Fe–g–C_3_N_4_ dependent Fenton-like photocatalysis is shown in Figure 6d. Fe doped g–C_3_N_4_, not only led to a soluble and reactive form of iron (Fe(III) and Fe(II)) at different pH values, but also the advantageously modified the redox properties by ligand-field effects [6]. When the Fe–g–C_3_N_4_ was excited by visible light, the electrons shifted to the conduction band and the holes left in its valence band (Equation (2)). As the interfacial charge transfer effect, parts of the photo-excited electrons reduced Fe(III) into Fe(II) (Equation (3)) (Fe(III)/Fe(II), 0.77 V, vs. NHE) [38]. Normally, the transformation from Fe(III) to Fe(II) is the decisive step for the whole reaction, much lower than the other steps. Nevertheless, it is facilitated with Fe–N ligands, thus created an excellent catalytic ability of Fe–g–C_3_N_4._ Then, Fe(II) reacted with H_2_O_2_ to produce •OH (Equation (4)), which oxidized RhB with a high efficiency. Moreover, the photo excited electron could also combine with the dissolved O_2_ in the solution to form•O_2_^−^ (Equation (5)).
(2)$$\mathrm{hv}+\to e^{-}+h^{+}$$
(3)$${\mathrm{Fe}}^{3+}+e^{-}\to {\mathrm{Fe}}^{2+}$$
(4)$${\mathrm{Fe}}^{2+}+H_{2}O_{2}\to {\mathrm{Fe}}^{3+}+\mathrm{OH}+{\mathrm{OH}}^{-}$$
(5)$$O_{2}+e^{-}\to O_{2}^{-}$$


## 4. Conclusions

In summary, the produced Fe–g–C_3_N_4_ by two-step calcination was successfully synthesized. Secondary calcination can increase specific surface area and enhance Fenton-like photocatalysis efficiency. When iron was doped into the g–C_3_N_4_ framework, Fe–N bonds were formed, which could effectively reduce the electron-hole recombination rate and broaden the scope of reaction pH in the range of 3–9. The main active oxygen species were •OH, followed by •O_2_^−^ and hole electrons. This produced catalyst of Fe–g–C_3_N_4_ shows excellent reusability and stability, and can be a promising candidate for dye degradation.

## Figures and Tables

**Figure 1 nanomaterials-10-00676-f001:**
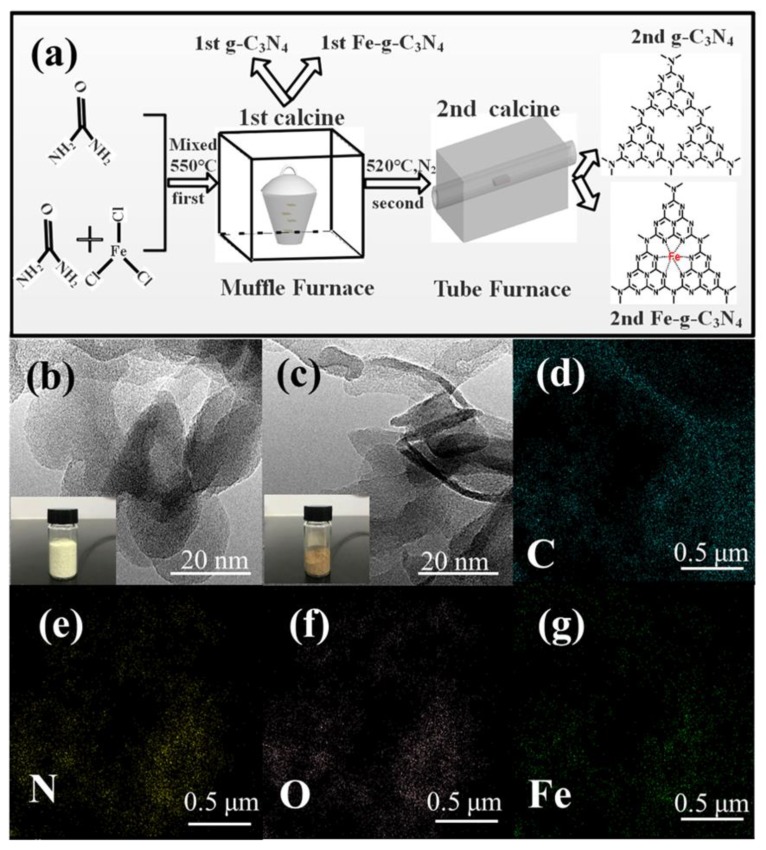
(**a**) Schematic of the one-step and two-step calcination process for the synthesis of g–C_3_N_4_ and Fe–g–C_3_N_4_, (**b**) TEM images of 2^nd^ g–C_3_N_4_, and (**c**) 2^nd^ Fe–g–C_3_N_4_; (**d**–**g**) element mapping of 2^nd^ Fe-g-C_3_N_4_.

**Figure 2 nanomaterials-10-00676-f002:**
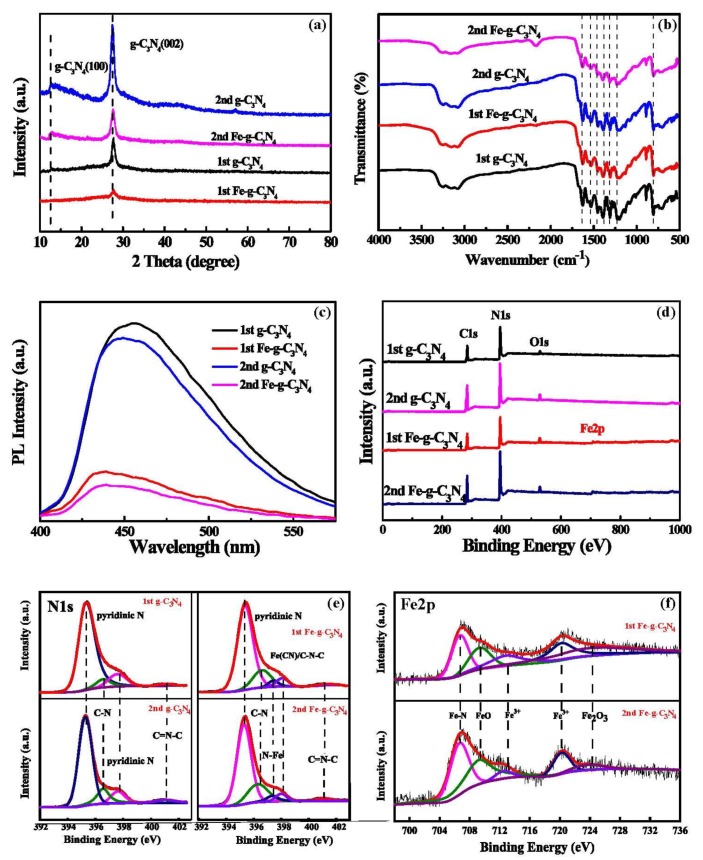
Characterization of one-and two-step g–C_3_N_4_ and Fe–g–C_3_N_4_ by XRD (**a**); FTIR (**b**), photoluminescence (**c**); and XPS (**d**–**f**) with zooms of the N1s (**e**) and Fe2p (**f**) spectra.

**Figure 3 nanomaterials-10-00676-f003:**
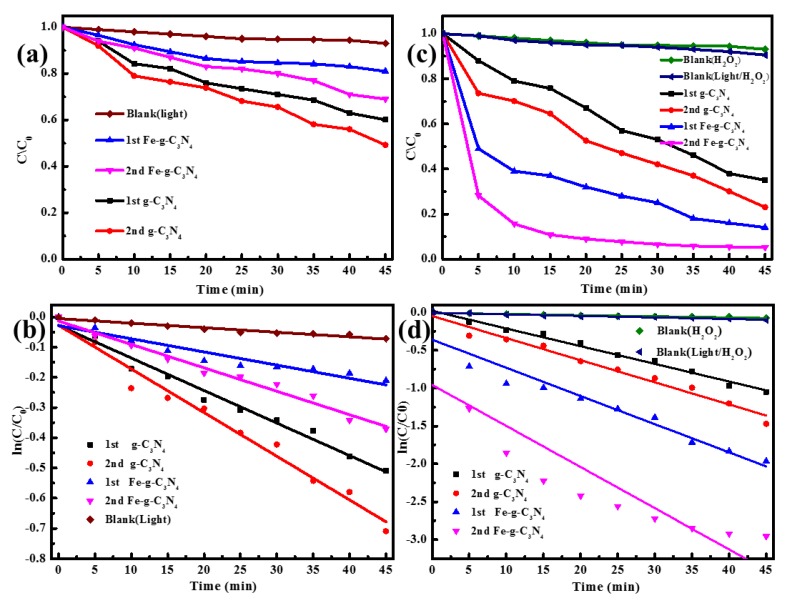
The photocatalysis performance and the pseudo-first-order kinetics fitted curves of RhB (**a**,**b**); and Fenton-like photocatalysis performance and the pseudo-first-order kinetics fitted curves of RhB (**c**,**d**).

**Figure 4 nanomaterials-10-00676-f004:**
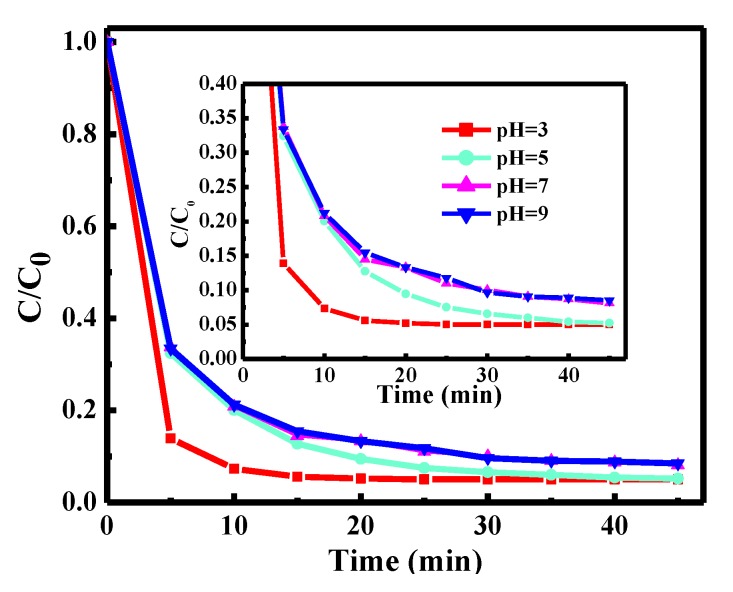
The different pH value of RhB degradation (2^nd^ Fe-g-C_3_N_4_ only). The solution pH was adjusted by 0.1 mol/L HCl or NaOH, and the pH change of the treated solution was small during the reactions (±0.2).

**Figure 5 nanomaterials-10-00676-f005:**
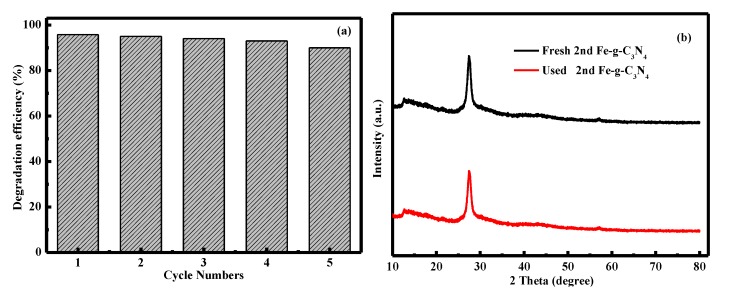
Reusability of the 2^nd^ Fe–g–C_3_N_4_ after five cycles (**a**); XRD patterns of the fresh and used 2^nd^ Fe–g–C_3_N_4_ (**b**).

**Figure 6 nanomaterials-10-00676-f006:**
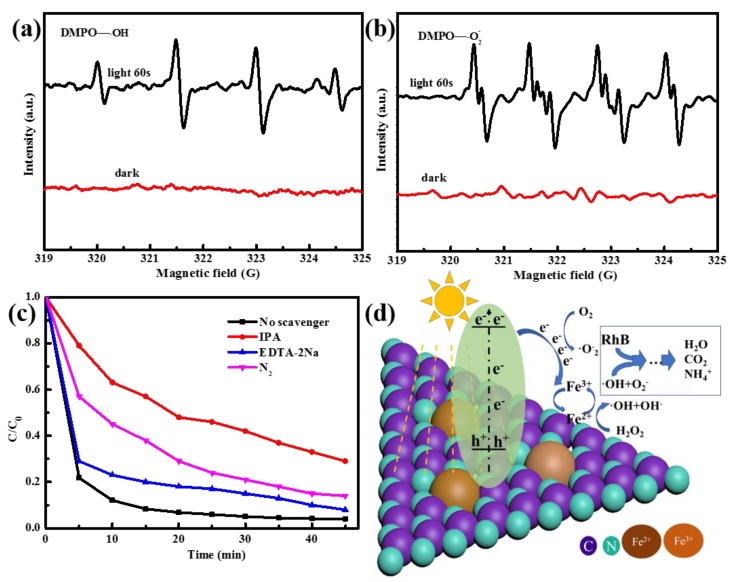
RhB degradation curves with different scavengers over 2^nd^ Fe–g–C_3_N_4_ (**a**), EPR spectra of DMPO/•OH (**b**), EPR spectra of DMPO/•O_2_^−^ (**c**), and a schematic illustration of mechanism (**d**).

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
