# Peer review of "Facile Production of a Fenton-Like Photocatalyst by Two-Step Calcination with a Broad pH Adaptability"

_nanomaterials, 2020, doi:10.3390/nano10040676_

Round 1

Reviewer 1 Report

Brief summary:

The paper studied the effect of Fe-doping and a two-step calcination on the photocatalytic properties of graphite carbon nitride (Fe-g-C3N4). The photocatalyst is synthesized and characterized using TEM, XRD and FT-IR methods. Its photocatalytic performance is investigated in photocatalytic degradation of rhodamine-B.

Broad comments:

The results are interesting but the paper is structured poorly and requires English language improvement. Introduction section is very short with minimum citations and references to the similar background studies. The paper includes a lot of images that are not discussed in detail. Conclusion section is not specific and focuses to the general principles of photocatalysis.

Specific comments:

Abstract – Line 14: 95% is the degradation efficiency not degradation rate that refers to the kinetic studies.

Abstract – Line 15-18: Discuss it in detail in the results and discussion section and include the representative chemical reactions and molecular structures of the species.

Introduction – (All) Literature review is very short with few citations of the past studies in this area. More citations are required to enhance the quality of the paper.

Line 32: Introduction: In the introduction section, include the type and examples of chemical contaminants that can be treated using this type of photocatalyst.

Line 42 – Please explain the novelty of the research.

Line 46- Materials and Methods: I’d suggest to include subsections (materials, preparation and characterization of the photocatalyst, analytical methods, and experimental design).

Line 51 – Is there any reference for the photocatalyst preparation method? If so, it needs to be cited. If there is no reference method, explain why this method is used.

Line 65 – Indicate the volume of the tested samples. Were they equal of different?

Line 68 – A blank sample without any photocatalyst instead of catalyst-negative blank.

Line 69 – Is it a UV-Vis Spectrophotometer? Include it in the text.

Line 71 – Detailed discussion of the results has to be added to this section.

Line 78 – What is the pore size and why is it important?

Line 94 – What does the fluorescent peak show?

Line 98 – Is it the chemical composition on the surface?

Line 99 – Fe (III) proportion to what?

Line 104 – How the proportion of total N is calculated?

Line 107 – How metallic Fe is detected and why it is mentioned in the text that it is not detected?

Line 112 – Degradation performance section: Adding degradation rate and kinetic of photocatalytic reactions as a basis of comparison increases the validity of the paper. Has it been studied before? If so, compare the degradation rate or performance with previous studies. Please also explain the Fenton like mechanism in detail including the chemical reactions.

Please also explain why changes in pH, changes the photocatalytic activity and also discuss how adding H2O2 improves the photocatalytic activity.

Line 113-120 – The graphs have to be explained in detail in the text.

Line 121 – Include chemical reactions here.

Line 124 - The production of reactive species (…) resulted in degradation of RhB.

Line 113 – Please quantify and use numbers instead of the word “better”.

Figure 3 – d: What are yellow balls? Species is not presented.

Line 134 – 135: Conclusion, The formed Fe-N complex….: Please explain why and how in section 3.2.

Line 132 – Conclusion: must be improved.

Figure S1: What catalyst is used? In the bar chart showing H2O2 concentration, is there any catalyst? If yes, include the amount.

Figure S3: What type of catalyst used here and why?

Reviewer 2 Report

The paper is well organized and has some novelty.

I recommend that the author investigate the adsorption capacity
and reusability of the catalysts and distinguish between
adsorption and decomposition of the RhB.
Authors investigate the pH affect. The pH was set before starting
phot-degradation but there is no information about the change of pH during the degradation of RhB.

Reviewer 3 Report

Nanomaterials-695024

report

Facile production of a Fenton-like photocatalyst by 2 two-step calcination with a broad pH adaptability

The authors report a Fe-doped graphitic carbon nitride (Fe-g-C3N4) composite obtained by a two-step calcination method. This composite is applied as catalyst in visible light for the degradation of rhodamine B (RhB) in the range of pH (3-9). The manuscript proposes a photocatalytic mechanism involving Fenton-like reaction between H2O2 and Fe (II) ion to generate hydroxyl radicals (•OH) for the degradation of RhB.The mechanism is not supported by the data in Figure 3.

The photoluminescence spectra show higher luminescence between 425-450nm (1st Fe-g-C3N4) and 430-480 nm (2st Fe-g-C3N4), however the spectra do not show the absorbance wavelength region for  Fe-g-C3N4, which is essential  for choosing the exposure light wavelength.

The conclusion "The reactive species (•OH, •O2- and holes) played roles in RhB degradation" is a general phrase  for any photocatalytic reaction and not particularly for RhB.

Round 2

Reviewer 3 Report

Report

nanomaterials-695024

Facile production of a Fenton-like photocatalyst by two-step calcination with broad pH adaptability

There are still inadvertent aspects relating to the photocatalytic effect of Fe-doped graphitic carbon nitride towards Rhodamine B oxidative degradation process and also to the pH region of action.

It is known that Rhodamine B decolorizes by H2O2:

"H2O2 can decolorize about 99% of rhodamine B after 1.5 h… "page 8 in: "Thao et al., Journal of Science: Advanced Materials and Devices 2 (2017) 317e325", also "AlHamedi et al., Desalination 239 (2009) 159–166","Daneshwar Desalinisation  230(1-3) (2008)16-26".This process of oxidation can be catalysed by Fe-doped graphitic carbon nitride. However, the data provided in the article do not demonstrate the assignments. How the pH was changed? Please add at the figure 4 caption this detail. Can be that the chemical for pH changes interacts with the catalyst and change the composition.

Also in the mechanisms of catalysis the Fe3+ is shown free however it is in a six-coordinated complex form [Fe(III)(N-...)6]3+, the reduction of Fe(III) to Fe(II) involves structural changes of the complex or inflexibility in reduction.

In the figure 2a, which are the bands assigned to the Fe(III) in the assumed Fe-doped graphitic carbon nitride? One cannot see additional peaks on the red and magenta XRD patterns (Fe-doped graphitic carbon nitride) compared to those marked with blue and black (graphitic carbon nitride).

The mechanism and the pH range must be reconsidered and better argued.
